# Barriers to Physical Activity for Women with Physical Disabilities: A Systematic Review

**DOI:** 10.3390/jfmk8020082

**Published:** 2023-06-16

**Authors:** Jurgi Olasagasti-Ibargoien, Arkaitz Castañeda-Babarro, Patxi León-Guereño, Naroa Uria-Olaizola

**Affiliations:** 1Health, Physical Activity and Sports Science Laboratory, Department of Physical Activity and Sports, Faculty of Education and Sport, University of Deusto, 48007 Bilbao, Spain; jurgi.olasagasti@deusto.es (J.O.-I.); arkaitz.castaneda@deusto.es (A.C.-B.); 2Deusto Sports and Society, Department of Physical Activity and Sports, Faculty of Education and Sport, University of Deusto, 48007 Bilbao, Spain; naroa.uria@deusto.es

**Keywords:** physical activity, barriers, women, physical disability

## Abstract

Physical activity is essential for women with physical disabilities. This review aims to identify the barriers they face in practicing sport. A systematic review was conducted using the PubMed/Medline, Scopus, and Web of Science databases in January 2023, with an update in March 2023. The eligibility criteria used for inclusion were as follows. (i) Women with physical disabilities; (ii) women who engage in or want to engage in physical activities and/or sport, both adapted and non-adapted; (iii) identification of women’s barriers to such practice; (iv) research articles; and (v) papers written in English and published in peer-reviewed journals. The exclusion were as follows. (i) Women with illness, injury or transient physical activity difficulties; (ii) mention of rehabilitative physical activity; and (iii) results showing no differentiation in barrier types by gender. This review identified different barriers, grouped into eight types according to the differentiating factor, thus showing that disable people’s participation in physical activity is directly related to some specific barriers which seem to differ according to their gender. Therefore, the success of participation in physical activities depends not only on the user’s concern, but also on an inclusive social environment.

## 1. Introduction

An estimated 1.3 billion people, or 16% of the world’s population, currently suffer from some form of disability [1]. About 1.5 billion people live with a physical, mental, sensory, or intellectual disability worldwide [2], of which women are the most affected, [3], making up 58.6% of the total. It is well known that regular physical activity is essential for maintaining good health and preventing disease [4,5,6]; the beneficial effects of exercise and physical activity on numerous aspects of health are now well known and generally accepted [7,8,9]. The practice of physical activity improves the physical, mental, and social state of the individual [10]. However, only 40% of women take part in the minimum recommended amount of physical activity [4,11], and in general, people with disabilities are in poorer health than the general population [2,12,13]. People with different disabilities such as cerebral palsy or spinal cord injury often face significant barriers to participating in physical activity [14,15,16], and in particular, women with physical disabilities often face unique and multiple challenges to practicing physical activity, including social, psychological, and physical barriers [17].

There is growing awareness of the importance of physical activity for people with physical disabilities, and as a result, there are several research studies on the benefits of physical activity for people with physical disabilities [18,19,20,21], and some suggestions for the adaptation of activities, e.g., for mothers with physical disabilities [22] or interventions for the assistance of people with a spinal cord injury [23]. Physical/sports activities for people with disabilities contribute to their functional independence, improve their physical condition, performance and physical capacity, favor the prevention and correction of deformities and postural defects, reduce stress, and improve self-confidence, emotional states, relationships with others, enjoyment and interest, among other things [24,25,26]. Within the Spanish population specifically, there is no study on the sporting habits of people with disabilities [27], although there are studies on the low levels of participation in sports practice [28], and on barriers to sport, both for different types of disabilities [14,15,16,29,30] and for people with physical disabilities in particular. However, the involvement of gender differences in these barriers is not appreciated [31,32,33,34,35,36,37]. However, when it comes to participation in physical activity (between women and men with disabilities), [14,16,20] women show poorer levels of participation. Differences gender-wise are found in the perception of barriers to physical activity according to gender [14]. Therefore, it is vitally important to highlight the needs of women with disabilities by identifying the barriers that have been identified in the literature and making this reality visible, in order to encourage participation in physical activity by women with disabilities.

Studies can be found in the literature that examine different types of barriers and their multiple classifications, such as social barriers, including discrimination and stigma; psychological barriers, including low levels of self-esteem and lack of motivation; and physical barriers, such as lack of access to adapted facilities and equipment [38,39,40]. Moreover, there exist intra-personal barriers, such as poor body image or fear of injury [41], and interpersonal barriers, such as lack of support or disapproval from others [12]. It is worth noting the existence of the differentiation in types of barriers according to gender [42]. Ways to overcome these barriers should be explored and the most effective strategies to increase the participation of women with physical disabilities in physical activity should be discussed.

The aim of this research is to raise awareness of the specific barriers faced by women with physical disabilities in relation to physical activity, and to ensure that they are addressed by management to the greatest extent of their responsibilities, in order to promote greater participation in physical activity. For these reasons, this article presents the results of a systematic review of the literature on barriers to physical activity for women with physical disabilities.

## 2. Materials and Methods

### 2.1. Literature Searching Strategies

The present research is a systematic review that seeks to identify the perceived barriers to women with physical disabilities who engage in or want to engage in physical activity. The research was conducted according the Preferred Reporting Items for Systematic Reviews and Meta-Analyses (PRISMA) guidelines [43]. This allowed an adequate structuring of the review, which answers the following question: what are the perceived barriers of women with physical disabilities who engage in or want to engage in physical activity, both adapted and non-adapted?

The following databases were used to conduct a structured search: PUBMED/MEDLINE, Web of Science (WOS) and Scopus. Using these high-quality databases, the search was completed without limitation to any specific year, and results were included up to and including 31 March 2023. Articles were retrieved from electronic databases using the following search strategy (Table 1): Barrier* AND “physical activit*” OR “adapted physical activit*” OR sport* AND disab* AND women OR woman, with these terms appearing in the title or abstract. Keywords were selected based on background reading. Articles considered relevant to this field of activity were obtained using the snowball strategy linked to this equation. In addition, all relevant studies were found by reviewing the titles and abstracts of articles in databases and the results of literature searches. These articles were considered potentially relevant and were analyzed for their compliance with the inclusion criteria for the final analysis. In addition, the reference sections of all articles found were examined, and all titles and abstracts obtained were cross-checked in order to detect possible duplication or lack of actual studies on the topic. Titles and abstracts were also selected for further full-text review. Two different authors searched for previous studies separately (J.O.-I. and N.U.-O.), and possible discrepancies were discussed with a third author (P.L.-G.).

### 2.2. Inclusion and Exclusion Criteria

No discrimination was made by country, race, or age, in order to obtain the most possible research. Articles published in English or Spanish and only research articles were filtered out.

The eligibility criteria used for inclusion were (i) women with physical disabilities; (ii) women who engage in or want to engage in physical activity and/or sport, both adapted and non-adapted; and (iii) the identification of barriers to women’s physical activity and/or sport. The exclusion criteria applied to the research were the following: (i) women with illnesses (i.e., those not identified as a disability, such as women with cancer), injuries or situations of difficulty for physical activity of a specific or transient duration, such as pregnant women; (ii) women engaging in rehabilitation physical activity, as they have access to physical therapy activities; and (iii) results not showing gender differentiation within the type of perceived barrier.

### 2.3. Quality Criteria

Two authors assessed quality in terms of the methodology used, along with any risk of bias (J.O.-I. and N.U.O.). Lack of consensus was submitted to third party assessment (P.L-G.), following the protocol for search and selection of studies [44]. All the following points of the protocol were carried out. (1) Whether the selected studies met the eligibility criteria or not was determined; (2) all the papers found were compared with each other to find out whether the studies overlapped or not; and (3) the selection process was then carried out. (A) The search results were combined by identifying the DOI of each article (if not available, one was assigned to each article). (B) All the searches were input in a database, and duplicates removed by DOI for a first screening. (C) For the second screening, studies were sorted by title, and publication data were compared to remove duplicates, taking into account the title, authors, journal, year, and number and/or volume of the study. (D) All the identified full texts were recruited for analysis. (E) The decision to include a paper in the study was made considering the favorable opinion of two researchers (J.O.-I. and N.U.-O.); if there was a difference in opinion, it is a third researcher (A.C.-B.) who decides with a third opinion (following point 4 of the protocol, an established process of inclusion criteria in the case of disagreement, and point 5, the reason for the exclusion of the excluded studies).

### 2.4. Data Processing

All the selected articles had to identify a study outcome that referred specifically to women. These outcomes were grouped according to the type of outcome, corresponding to a differentiating factors which were as a result of an exploratory factor analysis (EFA) that explained 56.5% of the total variance: personal, physical, psychological, leadership, media support, coaching role, economic, others’ attitudes, social, and cultural/religious support [17]. The barriers identified in each study were categorized and unified according to each grouping factor. In addition to the results, the profile of the subjects and the type of intervention carried out in these studies were added to obtain the results of each study.

## 3. Results

### 3.1. Search Process

Of the 1425 articles found after searching different databases, only 9 articles were identified that met all the inclusion criteria for the purposes of the systematic review, which are presented in the flowchart [45] (Figure 1). Of these 1425 articles, 1015 were removed as duplicates. Of the remaining 410 articles, 369 were eliminated after examination of titles or abstracts. Of the 41 full-text articles assessed for eligibility, a further 33 papers were discarded because they did not identify gender in the type of barrier (*n* = 8), did not specify the type of barrier (*n* = 13), did not specify the type of disability of the subjects (*n* = 3), were systematic reviews of another topic (*n* = 5), or were not related to the defined object of study (*n* = 4). On the other hand, one article was found after the citation search [39]. Thus, the present systematic review included nine studies [39,41,46,47,48,49,50,51,52].

### 3.2. Differentiating Factor and Characteristics of Barriers

The different types of barriers identified as differentiating factors were based on the work of Bakhtiary et al.; the characteristics detected in each of them and the studies in which these barriers are mentioned are shown in Table 2.

Table 2 shows that there is no study that confirms that there are barriers according to the differentiating factors of media support and cultural/religious support.

Table 3 shows all the studies found with the most complete data on the study subjects, the intervention methodology used, and the types of barriers identified according to the corresponding factor. The most frequently mentioned factor as a barrier was the psychological factor (*n* = 7), with a lack of motivation predominating. This was followed by the management (*n* = 6) of the offers or possibilities to practice sport for women with physical disabilities, and the lack of social support (*n* = 6), understood as the lack of support both from the immediate environment and from other people in society for access to or the possibility of engaging in physical activity.

## 4. Discussion

The current review aims to identify the perceived barriers of women with disabilities to engaging in physical activity or sport. We did not find any existing studies with such objectives. Therefore, the main objective has been to identify several types of very complex barriers that cover several aspects through which they can be approached; we propose that knowledge of these will help professionals to guide and favor women with physical disabilities in accessing sports or physical activity, as happens in populations with other physical needs [53]. A total of eight types of barriers (personal, physical, psychological, direction, coach’s role, economic, others’ attitudes, and social support) were identified from nine studies.

This research has yielded many significant insights. First, the results demonstrate that the barriers to physical activity for women with physical disabilities are multiple and complex, and span multiple dimensions. Many women with physical disabilities find it difficult to be physically active [54]; this study has revealed that personal barriers such as age, fatigue, loneliness, lifestyle, or simply being a woman may limit participation in any physical activity. All these factors are responsible for the low participation of women with physical disabilities in sport, since women have been shown to be less likely to participate in sports than men [16].

There is sufficient theoretical evidence on the benefits of sporting activity [55]; in the case of people with disabilities, it contributes to their functional independence, improves their physical condition, performance and physical capacity, favors the prevention and correction of deformities and postural defects, reduces stress, improves self-confidence, emotional state, relationships with others, and enjoyment and interest, among other things [26,56,57,58]. However, a second important finding of this review is that barriers related to physical disability, such as health, mobility, or the degree of dependence on others, also prevent women from practicing sport, thus reducing the possibility of appropriating all the benefits mentioned above.

Another barrier to participation is often linked to psychological factors. Within the psychological variables, motivation is one of the best known, although it is well known that motivation can be influenced by different factors [59] and is considered the most prominent factor in adherence to physical activity [13,60], The same factors were found in this study, wherein women with disabilities experience different situations that lead to a lack of motivation being a barrier to practicing sport [41,46,47,49,50,52]. In addition to motivation, other psychological factors, such as fear [41,50,52], the perception of not being able to engage in physical activity [50], or a negative self-perception [39], form a barrier for women with physical disabilities.

Such women should have opportunities for physical activity in a safe and adapted environment in accordance with their needs, under the Sport Law 2007 [61]; moreover, society as a whole must be committed to making this a reality [62]. However, this study has shown that these people encounter many management barriers, such as poor accessibility or lack of adaptations to sports centers, lack of transport to sports facilities, lack of co-communication between professionals, and poor organizational management. Confined spaces and equipment that does not easily accommodate mobility limitations also prevent them from making full use of exercise equipment and space [16], and the lack of communication between different professionals is also often one of the main barriers [63] that leads to a lack of opportunities [64]. Therefore, we believe that it is necessary to theorize about the structure of sporting institutions, starting with an inclusive atmosphere that can then be implemented into the wider sporting culture and activities, in which all people belonging to the community can be participants, as also pointed out in another study [62].

The role of a coach is essential for the proper performance of physical activities to ensure the safety of the participant and the benefits of physical activity and sport [65]; the important role of trainers is evident [15]. Therefore, health and physical activity professionals should consider individuals’ abilities, needs, limitations, values, personality types and aptitudes in order to customize an adapted program and minimize the effects of possible barriers [53,66]. This study shows that training staff lack training in adapting physical activity or programs to the needs of users. There is a great lack of knowledge about the different sports modalities available for people with disabilities, especially those that are specific (e.g., boccia, slalom and goalball) [67]. Therefore, we believe that personalized attention may be the key to increasing levels of participation in sports, and to keeping programs challenging and attractive for people with physical disabilities. In addition, there should be a variety of offers, so that the user can select a program that best suits his or her needs [64,67].

It is clear that for women with physical disabilities to benefit from sport requires an approach that involves a wider range of programs. However, the economic context of the users must also be considered, as this barrier can have a crucial impact on participation in sport [35]. Taking into account that resources for this group are limited [31], when the financial costs of sporting activities increase, a barrier to participation is formed. In this sense, being a woman can also be an added barrier, as it can be more difficult to obtain sporting sponsors [68]. It is clear that for women with physical disabilities to benefit from sport, an approach that involves a wider range of programs is required. However, the economic context of the users must also be considered, as this barrier can have a crucial impact on participation in sport. Among the benefits of sports practice mentioned above, the improvement of relationships with others is defined by the following [21,26]; however, we should take into account the differentiation between social support in relationships with non-disabled people and in relationships with people with disabilities. Studies have shown an improvement in happiness as a result of perceived social support among people with disabilities [33]; however, this same improvement is not felt as a result of support from people without disabilities [46,47,49,51]. In addition to this, we would have to differentiate between social support from family and from acquaintances and other people. Family support [69] and the support of partners is fundamental, although barriers may exist [41,49,50]; however, the support of society and institutions also has an impact on successful sports practice [14,47,49]. Users are therefore more likely to show reduced adherence to exercise without this support [70,71]. It should also be noted that there are differences in social support according to gender [72]; in one study, it was found that men received more support from family and friends than women [73], just as in this study, wherein a lack of social support can be observed as a barrier to physical activity. Women’s participation in physical activity was shown to be lower than the participation of men with disabilities [14,16,20]; these differences are also significant when it comes to the perception of barriers to physical activity, according to gender [14], between men and women.

It is also worth noting that another barrier identified was social attitudes towards people with disabilities. Negative attitudes may become barriers to the full realization of human potential [38], and because physical disabilities are visible to others, they can lead to stigmatizing social experiences [53,74]. It is therefore necessary to eliminate disability stigma, as people with disabilities who perceive stereotyping to a high degree also perceive a lower quality of life [33,75]. A society that does not recognize and value people with functional diversity loses all the potential they have to offer [76]. Other barriers identified as differentiating factors were culture and media [17], but in this research, they were far less represented.

### Limitations

As this is a systematic review, our limitations are related to the studies inserted here. Although the selected sample is composed of a specific population (women with disabilities who want to perform or perform physical activity or sport), it is heterogeneous in terms of age, the type of physical limitation within the physical disability, and the context of the physical activity or sport performed by the women in the studies.

## 5. Conclusions

This review has enabled the identification of the barriers that women with physical disabilities encounter when performing physical activities. These findings highlight that participation in sports practice does not only depend on the participant. Society must know the barriers that women with physical disabilities face in practicing sports; this review publicizes those barriers in order to improve our best practices. Understanding and raising awareness of these barriers will allow interventions to be adapted to address the barriers, and to provide more targeted support and guidance to women with physical disabilities. As people with physical disabilities themselves have demonstrated and demonstrate every day, we must not forget that they are people with great abilities who want to participate, on equal terms, in physical activity. As active members of the society to which they belong, they have the right to live as independently as possible and with the highest possible quality of life. A society that does not recognize the value of people with functional diversity will lose all the potential they have to offer.

## Figures and Tables

**Figure 1 jfmk-08-00082-f001:**
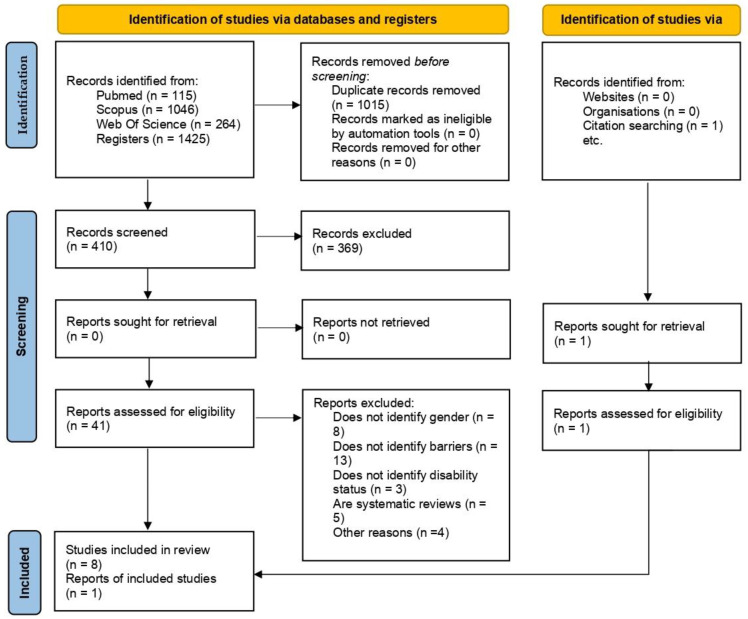
PRISMA flow diagram for study selection.

**Table 1 jfmk-08-00082-t001:** Search strategy.

	Articles Found		
	WOS	PUBMED	SCOPUS
1- Barrier* AND physical activit* AND disab* AND women	86	49	361
2- Barrier* AND physical activit* AND disab* AND woman	86	4	361
3- Barrier* AND adapted physical activit* AND disab* AND wom*	8	0	18
4- Barrier* AND sport* AND disab* AND wom* (All Fields)	21	14	71
5- Barrier* AND “physical activity” AND disab* AND wom*	63	48	235
Total without duplicates	97	55	397
	410		

**Table 2 jfmk-08-00082-t002:** Studies included in the systematic review: characteristics of barriers and relevant studios.

Differentiating Factor	Characteristics	Number of Studies	References
Personal	Age	1 study	[50]
	Solitude	1 study	[50]
	Lifestyle	2 studies	[50,52]
	Being a woman	1 study	[50]
	Lack of strategy	1 study	[50]
	Fatigue	1 study	[47]
Physical	Health	3 studies	[41,50,52]
	Mobility	2 studies	[47,50]
	Self-care	1 study	[50]
	Dependence	1 study	[50]
Psychological	Self-image	1 study	[39]
	Motivation	6 studies	[41,46,47,49,50,52]
	Fear	3 studies	[41,50,52]
	Perception of (in)capacity	1 study	[50]
Managerial	Accessibility or adaptations	4 studies	[47,48,49,50,51]
	Transport	4 studies	[47,49,50,52]
	Communication	4 studies	[47,49,50,52]
	Political organization	2 studies	[49,50]
Coach role	Lack of training	2 studies	[49,51]
Economic	High cost	3 studies	[47,49,52]
	Assistants’ expenses	1 study	[48]
Others attitudes	Looks	2 studies	[50,51]
Social support	Family	2 studies	[49,50]
	Colleague	2 studies	[41,50]
	Society	4 studies	[46,47,49,51]
	Institution	1 study	[49]
Cultural/religious support			
The media			

**Table 3 jfmk-08-00082-t003:** Data extraction and synthesis.

Author/s Year	Population	Intervention	Barriers Factor
Anderson et al., 2008 [41]	Subjects: 22 women (10–18 years): Type of injury: Cerebral palsy (7) Spina bifida (8) Osteogenesis (2) Amputee (2) Reduced mobility (2) Cerebral anoxia(1)	Semi-structured interviews	1, 3, 9
Cardenas et al., 2021 [49]	Subjects: 49 women	Questionnaire	1, 3, 4, 6, 7, 8, 9
Dlugonski et al., 2012 [46]	Subjects: 11 women (42.9 years ± 10.2) Type of injury: Multiple Sclerosis	Semi-structured interviews	3, 9
Henderson and Bedini, 1995 [47]	Subjects: 16 women (29–53 years): Type of injury: Rheumatoid arthritis (3) Multiple sclerosis (3) Spinal cord injury (3) Cerebral palsy (2) others (5)	In depth interviews	1, 2, 3, 4, 7, 9
Odette et al., 2003 [48]	Subjects: 45 women (+18 years) (mean 43, 18–80 years) Type of injury: Rheumatological Neurological Musculoskeletal disorders	Focus group	4, 7
Rauch et al., 2013 [50]	Subjects: 13 women (44, 31–75 years)	Focus group and individual interviews	1, 2, 3, 4, 5, 8, 9
Richardos et al., 2017a [51]	Subjects: 8 women (+18 years), (mean 40, 23–60 years)	Focus group and individual interviews	8, 9
Richardos et al., 2017b [39]	Subjects: 8 women (+18 years), (mean 43 years ± 13)	Semi-structured interviews, videoconferences, telephone interviews	3, 4, 6
Rimmer et al., 2000 [52]	Subjects: 53 (18–64 years) (4%, 18–34 years; 31%, 35–49 years and 65%, 50–64 years) Type of injury: Arthritis (30) Apoplexy (22) Multiple sclerosis (14) Diabetes (10) Lung diseases (8)	Telephone interviews	1, 2, 3, 4, 7

1. Personal; 2. physical; 3. psychological; 4. managerial; 5. the media; 6. coach’s role; 7. economic; 8. others’ attitudes; 9. social support; 10. cultural/religious.

## Data Availability

Not applicable.

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
