# Peer review of "Barriers to Physical Activity for Women with Physical Disabilities: A Systematic Review"

_jfmk, 2023, doi:10.3390/jfmk8020082_

Round 1

Reviewer 1 Report

Points to review:

- review the undifferentiated use of 'physical activity’ (for example, in the title), 'physical exercise' (in the abstract and objectives l.64-68), and 'sport' (abstract, methods l.74). It is confusing for the reader;

-what is the relevance of mentioning information about the context of Spain in the introduction? (e.g., l.29, 50). The information is confusing for the reader…

- l.46 - 50, the statement deserves more references as there is plenty of literature on the subject;

- female gender? Change to woman/women

- the choice of an a priori categorization based on what the authors called 'differentiating factors' should be well justified and not just refer to a study;

- in the presentation of the results, there is no clarification about the interpretation process that led to the categorization of the results of the studies into the different categories ('differentiating factors'); the authors do not clarify or legitimize this process for the reader;

- in the discussion, the focus should be on barriers for disabled women practitioners, but only 12 lines refer to this (l.192-196; l.209-213; l.215-217); - some statements should be revised (e.g., l.280-282 - the information "can be grouped" does not seem correct because the categories did not emerge from the data. The methodological option was to analyze the studies based on categories defined a priori, the 'differentiating factors.'

- It is not stated whether other categories other themes emerged from the analysis of the studies other than those categorized in the 'differentiating factors';

- review the references (e.g., 48)

Author Response

The authors appreciate the time you devoted to reading our manuscript and helping us to craft an improved version of the investigation. We are pleased to clarify your concerns, which we believe have improved the quality and applicability of this work. Please, find below our responses to each of your observations. We have made a concerted attempt to systematically address the specific concerns raised for this revision and we have highlighted the alterations to this revision within the manuscript with track changes for your convenience.

Reviewer reports:

Reviewer 1:

Background

REVIEWER 1: review the undifferentiated use of 'physical activity’ (for example, in the title), 'physical exercise' (in the abstract and objectives l.64-68), and 'sport' (abstract, methods l.74). It is confusing for the reader;

AUTHORS: Thank you for your comment. We changed physical exercise into “physical activity” throughout the manuscript. Lines: 12, 35, 235, 282 and 293.

AUTHORS: Other than that, as you suggested, we deleted or changed “sport practice” into “physical activity” in the cases that we considered that were confusing. Lines: 41, 76, 80,

REVIEWER 1: what is the relevance of mentioning information about the context of Spain in the introduction? (e.g., l.29, 50). The information is confusing for the reader…

AUTHORS: Thank you for your comment. As you suggested we used a reference which gets a global view of the issue, instead of focusing in a certain social context like Spain.

Line 30: “Martin Ginis, K. A., et al. "Participation of people living with disabilities in physical activity: a global perspective." The Lancet 398.10298 (2021): 443-455”.

REVIEWER 1: l.46 - 50, the statement deserves more references as there is plenty of literature on the subject;

AUTHORS: Thank you for your comment. As you suggested we added the following references to endorse the statement:

Line: 53, 54.

“Arbour, Kelly P., et al. "Moving beyond the stigma: The impression formation benefits of exercise for individuals with a physical disability." Adapted Physical Activity Quarterly 24.2 (2007): 144-159.https://doi.org/10.1123/apaq.24.2.144”

“Asonitou, Katerina, et al. "Effects of an adapted physical activity program on physical fitness of adults with intellectual disabilities." Advances in Physical Education 8.3 (2018): 321-336. 10.4236/ape.2018.83028”  

 “Martin Ginis, K. A., et al. "Participation of people living with disabilities in physical activity: a global perspective." The Lancet 398.10298 (2021): 443-455”. https://doi.org/10.1016/S0140-6736(21)01164-8

REVIEWER 1: female gender? Change to woman/women

AUTHORS: Thank you for your comment. We changed female gender by woman or women. Lines: 131,132.

In line 31 we deleted “as a social classification of the female gender” since otherwise it was redundant.

REVIEWER 1:       the choice of an a priori categorization based on what the authors called 'differentiating factors' should be well justified and not just refer to a study;

AUTHORS: Thank you for your comment. As you suggested we explained and justified the called “differentiating factors”.

Lines: 133, 134. “s, which were as a result of an exploratory factor analysis (EFA), that explained 56.5% of the total variance”

REVIEWER 1: in the presentation of the results, there is no clarification about the interpretation process that led to the categorization of the results of the studies into the different categories ('differentiating factors'); the authors do not clarify or legitimize this process for the reader

AUTHORS: Thank you for your comment. As you suggested in the previous review, we added some specific information related to the different categories of “differentiating factors”. Lines: 133, 134. “s, which were as a result of an exploratory factor analysis (EFA), that explained 56.5% of the total variance”

AUTHORS: When it comes to the interpretation of the process that led to the categorization, we added the reference of the article in which the factors were described. Lines: 159,160. “based on Bakhtiary et al. (18)”

REVIEWER 1: in the discussion, the focus should be on barriers for disabled women practitioners, but only 12 lines refer to this (l.192-196; l.209-213; l.215-217); - some statements should be revised (e.g., l.280-282 - the information "can be grouped" does not seem correct because the categories did not emerge from the data. The methodological option was to analyze the studies based on categories defined a priori, the 'differentiating factors.'

AUTHORS: Thank you for your comment. As you suggested, we revised the discussion section paying special attention to those sentences that you suggested. Line 202: “, since women showed to be less likely to participate in sports than men”.

Other than that, as the categories were not emerged from the data, we decided to delete this sentence. Lines 287-289. “where most of which can be grouped into eight categories: personal, physical, psychological, management or organizational, coach, economic, attitudinal, and social support barriers”. When it comes to the methodological option to analyze the studies based on the “differentiating factors” show line number of the original manuscript in which those factors are describing previous research: Lines: 193. Personal Barriers; Lines: 202. Physical Barriers; Lines: 206. Psychological barriers; Lines: 218. Management Barriers; Lines: 228. Role of coach; Lines: 242. Economic context; Lines: 254. Social Support; Lines: 264 Social Attitudes

REVIEWER 1: It is not stated whether other categories other themes emerged from the analysis of the studies other than those categorized in the 'differentiating factors'

AUTHORS: Thank you for your comment. We are not quite sure about the question. We would appreciate a wider clarification, thank you.

REVIEWER 1: review the references (e.g., 48)

AUTHORS: Thank you for your comment. We revised the references, included the number 48. Line 446:

“Odette F, Israel P, Li A, Ullman D, Colontonio A, Maclean H, et al. Barriers to wellness activities for Canadian women with physical disabilities. Health Care Women Int [Internet]. 2003 Feb;24(2):125–34. Available from: http://www.tandfonline.com/doi/abs/10.1080/07399330390170105”

Reviewer 2 Report

JFMK-2397022 presents a review paper for barriers to PA for women with disabilities. While some parts of this paper were interesting, other areas could be improved. I hope the authors consider my feedback.

·         Lines 27-28: Please delete this sentence.  

·         Line 29: Define INE before use.

·         Line 35: Be careful about using “physical exercise” and “physical activity” interchangeably. Maybe just stick to “physical activity” throughout.

·         Line 38: Which disability types exactly?

·         Line 45: Clean typos here and throughout “o”.

·         Lines 83-84: How were these keywords selected? Please state in the text.

·         Line 209: Again, typo “(15)(60)”.

·         Discussion: There needs to be a paragraph of text devoted to the differences observed in men vs. women for this section. Line 261 hints at these differences, but more pronunciation is needed.

·         The abstract should be more focused on the findings and less on the methods. For example, lines 13-19 are methods-based. Please revise.

·         Make any changes to the abstract that align with those from the text.

Author Response

Reviewer 2:

REVIEWER 2: JFMK-2397022 presents a review paper for barriers to PA for women with disabilities. While some parts of this paper were interesting, other areas could be improved. I hope the authors consider my feedback.

AUTHORS: Thank you for your comment. We will try to follow step by step your suggestions to improve the quality of our manuscript.

REVIEWER 2: Lines 27-28: Please delete this sentence.  

AUTHORS: Thank you for your suggestion. We deleted that sentence from lines 27-28 and also the reference connected to this sentence from the bibliographic section.

REVIEWER 2: Line 29: Define INE before use.

AUTHORS: Thank you for your comment. As another reviewer suggested a change related to this source, we decided to include a worldwide perspective data, and therefore we deleted the sentence where INE was specified. Lines: 30,32 “In Spain according to INE data from 2020 (2) approximately 4 million people claim to have some form of disability”. And we added instead the following sentence in Lines: 29,30”

REVIEWER 2: Line 35: Be careful about using “physical exercise” and “physical activity” interchangeably. Maybe just stick to “physical activity” throughout.

AUTHORS: Thank you for your comment. We changed physical exercise into “physical activity” throughout the manuscript. Lines: 12, 35, 235, 282 and 293.

REVIEWER 2: Line 38: Which disability types exactly?

AUTHORS: Thank you for your comment. We specified some examples of different disabilities. Lines 40: “like cerebral palsy or spinal cord injury”

REVIEWER 2: Line 45: Clean typos here and throughout “o”.

AUTHORS: Thank you for your comment. We corrected the error.

Line 48: disabilities (24) “or” interventions

REVIEWER 2: Lines 83-84: How were these keywords selected? Please state in the text.

AUTHORS: Thank you for your comment. As you suggested we added a sentence explaining how were these keywords selected. Line 89: “Keywords were selected based on background reading”.

REVIEWER 2: Line 209: Again, typo “(15)(60)”.

AUTHORS: Thank you for your comment. We corrected the error.

Line 215: physical activity “(15,60)”

REVIEWER 2: Discussion: There needs to be a paragraph of text devoted to the differences observed in men vs. women for this section. Line 261 hints at these differences, but more pronunciation is needed.

AUTHORS: Thank you for your comment. We added a paragraph emphasizing the differences according to the gender when it comes their participation in physical activity

Lines: 277-280. “Women’ participation in physical activity showed to be lower than in men with disabilities (16,18,22) being also these differences significant when it comes to the perception of barriers to physical activity according to gender (16) between men and women.

REVIEWER 2: The abstract should be more focused on the findings and less on the methods. For example, lines 13-19 are methods-based. Please revise.

AUTHORS: Thank you for your comment. As It is a systematic review, we considered important to properly define the method, however, we added some more finding in order to complete better this section. Line: 21-23. “, showing thus that disable people’ participation in physical activity is directly related to some specific barriers , which seems to differ depending on the gender. Therefore, t”

REVIEWER 2: Make any changes to the abstract that align with those from the text.

AUTHORS: Thank you for your comment. We tried to align the abstract with the text with this sentence: Line: 21-23. “, showing thus that disable people’ participation in physical activity is directly related to some specific barriers, which seems to differ depending on the gender. Therefore, t”

Reviewer 3 Report

This study contains important goals for women with physical disabilities to facilitate physical activity. To make this review more meaningful, the authors should re-consider the following and revise the manuscript.

Gender differences should be considered in more depth. This review only includes one article on the results for men, but whether there were gender differences in the interpretation of the results of this review is a crucial point. It may also be helpful to include previous research on men (e.g., what is known for men but not for women) in the introduction to clarify this discussion.

Author Response

The authors appreciate the time you devoted to reading our manuscript and helping us to craft an improved version of the investigation. We are pleased to clarify your concerns, which we believe have improved the quality and applicability of this work. Please, find below our responses to each of your observations. We have made a concerted attempt to systematically address the specific concerns raised for this revision and we have highlighted the alterations to this revision within the manuscript with track changes for your convenience.

Reviewer 3:

REVIEWER 3: This study contains important goals for women with physical disabilities to facilitate physical activity. To make this review more meaningful, the authors should re-consider the following and revise the manuscript.

AUTHORS: Thank you for your comment. We will consider your suggestions and revise the manuscript.

REVIEWER 3: Gender differences should be considered in more depth. This review only includes one article on the results for men, but whether there were gender differences in the interpretation of the results of this review is a crucial point. It may also be helpful to include previous research on men (e.g., what is known for men but not for women) in the introduction to clarify this discussion.

AUTHORS: Thank you for your comment. We wrote a sentence referring to gender difference at the introduction section, in order to . Lines:57-63. “However, when it comes to participation in physical activity between women and men with disabilities (16,18,22) women show a poorer participation. Differences genderwise are found in the perception of barriers to physical activity according to gender (16).  Therefore, it is vitally important to highlight the needs of women with disabilities by identifying the barriers that have been identified in the literature and making this reality visible in order to encourage participation in physical activity by women with disabilities”.

Round 2

Reviewer 2 Report

The authors did a nice job addressing my previous concerns. A couple comments:

***Lines 54-55: These in-text citations are not correctly formatted.

***The References section is not formatted. For example, there is no Reference 2 or 3. There are also no numbers for citations near Reference 26. Please format. 

Author Response

The authors appreciate the time you devoted to reading our manuscript and helping us to craft an improved version of the investigation. We are pleased to clarify your concerns, which we believe have improved the quality and applicability of this work. Please, find below our responses to each of your observations. We have made a concerted attempt to systematically address the specific concerns raised for this revision and we have highlighted the alterations to this revision within the manuscript with track changes for your convenience.

Reviewer reports:

Reviewer 2:

REVIEWER 2: Lines 54-55: These in-text citations are not correctly formatted.

AUTHORS: Thank you for your comment. We formatted correctly the references and we changed the rest of the numbers (references according to these changes). For E.g., reference 4 becomes 3, and so on until reference 12 becomes 11. Reference 13 is deleted since is the same as reference 2. From that number on, 14 becomes 12, and so on until number 25, which is 23, then we added the new two references, 24 and 25, and therefore, reference 26 and until the end remain the same, until 76.

REVIEWER 2: The References section is not formatted. For example, there is no Reference 2 or 3. There are also no numbers for citations near Reference 26. Please format. 

AUTHORS: Thank you for your comment. We formatted the reference section including the new references and changed the number of the other references according to these changes.
